# Identification and Characterization of the MIKC-Type MADS-Box Gene Family in *Brassica napus* and Its Role in Floral Transition

**DOI:** 10.3390/ijms23084289

**Published:** 2022-04-13

**Authors:** Enqiang Zhou, Yin Zhang, Huadong Wang, Zhibo Jia, Xuejun Wang, Jing Wen, Jinxiong Shen, Tingdong Fu, Bin Yi

**Affiliations:** 1National Key Laboratory of Crop Genetic Improvement, National Center of Rapeseed Improvement, College of Plant Science and Technology, Huazhong Agricultural University, Wuhan 430070, China; zhouenqiang0526@163.com (E.Z.); wanghuadong_jxau@163.com (H.W.); zhibojia@webmail.hzau.edu.cn (Z.J.); wenjing@mail.hzau.edu.cn (J.W.); jxshen@mail.hzau.edu.cn (J.S.); futing@mail.hzau.edu.cn (T.F.); 2Jiangsu Yanjiang Institute of Agricultural Sciences, Nantong 226001, China; zhangyin19940203@163.com (Y.Z.); wangxj4002@163.com (X.W.)

**Keywords:** *Brassica napus*, CRISPR/Cas9, MIKC-type MADS-box, flowering, *BnaSVP*, *BnaSEP1*

## Abstract

Increasing rapeseed yield has always been a primary goal of rapeseed research and breeding. However, flowering time is a prerequisite for stable rapeseed yield and determines its adaptability to ecological regions. MIKC-type MADS-box (MICK) genes are a class of transcription factors that are involved in various physiological and developmental processes in plants. To understand their role in floral transition-related pathways, a genome-wide screening was conducted with *Brassica napus* (*B. napus*), which revealed 172 members. Using previous data from a genome-wide association analysis of flowering traits, *BnaSVP* and *BnaSEP1* were identified as candidate flowering genes. Therefore, we used the CRISPR/Cas9 system to verify the function of *BnaSVP* and *BnaSEP1* in *B. napus*. T0 plants were edited efficiently at the *BnaSVP* and *BnaSEP1* target sites to generate homozygous and heterozygous mutants with most mutations stably inherited by the next generation. Notably, the mutant only showed the early flowering phenotype when all homologous copies of *BnaSVP* were edited, indicating functional redundancy between homologous copies. However, no changes in flowering were observed in the *BnaSEP1* mutant. Quantitative analysis of the pathway-related genes in the *BnaSVP* mutant revealed the upregulation of *SUPPRESSOR OF OVEREXPRESSION OF CONSTANS 1* (*SOC1*) and *FLOWERING LOCUS T* (*FT*) genes, which promoted early flowering in the mutant. In summary, our study created early flowering mutants, which provided valuable resources for early maturing breeding, and provided a new method for improving polyploid crops.

## 1. Introduction

Flowering is important for plant development and indicates the end of vegetative growth; it is a result of developmental events and the one of basic requirements for reproduction. It is also an important agronomic trait in breeding [1]. Environmental conditions such as temperature, photoperiod, and diurnal rhythm changes, as well as internal hormone levels, can affect flowering. External and internal factors coordinate and interact with each other and affect flowering [2,3]. Plants have evolved multiple mechanisms to cope with the environment, such as vernalization and circadian rhythms. Furthermore, plants have been shown to adjust their flowering time in response to an adverse environment [4,5]. The study of flowering-related genes is of great significance for crop breeding and agricultural production.

Transcription factors (TFs) are major drivers of evolution and domestication that can be exploited for crop improvement [6]. MADS-box TFs are one of the largest gene families found in plants [7]. MADS-box gene families are classified into type I and type II, based on their gene characteristics [8]. Compared with the type I MADS-box gene, the type II MADS gene function has been fully studied, and most of these genes can regulate flowering time and flower development. The type II MADS-domain protein contains four types of domains: MADS, K, I, and C-terminal [9]. The MADS-domain enables the DNA binding, nuclear localization, and dimerization of TFs [10,11]. Notably, the K- and C-terminal domains are thought to mediate the interaction between proteins [12], while the I domain forms DNA-binding dimers with low conservation [13]. According to the structural characteristics of type II, it is also called MIKC [12]. The MIKC type has been subdivided into MIKC^C^ and MIKC* groups, and MIKC^C^ contains 13 different gene subfamilies [14,15,16].

In recent years, the MIKC family has been identified in many crops, such as *Prunus mume* [17], grapevine [14], *Prunus persica* [18], and soybean [19]. Furthermore, evolution and functional analyses of *A. thaliana* [20], as well as other important crops, including banana, rice, and *B**. rapa*, have been carried out [21,22,23]. Members of the MIKC family not only include ABCDE floral development model genes, but also a large number of flowering genes, such as *FLOWERING LOCUS C* (*FLC*) and *SHORT VEGETATIVE PHASE* (*SVP*) [15]. Ren et al. analyzed the MIKC family in cotton and identified 110 genes. RNA sequencing revealed that most MIKC genes were highly correlated with flowering, and no *FLC* subclass genes were present in cotton. In addition, the overexpression of *GhAGL17.9* significantly increased the expression levels of *CONSTANS* (*CO*), *LEAFY* (*LFY*), and *SOC1*, and induced the early flowering of plants [24]. MIKC genes have been studied in wheat for nearly twenty years. The most prominent example of such a gene is *VERNALIZATION1* (*VRN1*) [25,26,27]. A functional analysis of the MIKC family genes revealed that multiple genes could regulate the flowering time of crops, including *FLC*, *SVP*, and *FLOWERING LOCUS M* (*FLM*), which could delay flowering [28,29,30], whereas *SOC1*, *STRESS ENHANCED PROTEIN*-like (*SEP*-like), *APETALA1* (*AP1*), *FRUITFUL* (*FUL*), and *AGAMOUS-LIKE 24* (*AGL24*) could promote flowering [31,32,33,34,35]. The function of the MIKC gene involves many aspects, including roots, flowers, and seeds [36]. Therefore, understanding the role of the MIKC gene has great potential and can provide a basis for crop breeding and improvement [37].

Genome-wide association study (GWAS) is an effective analytical method developed in recent years to analyze the genetic basis of crop phenotypic diversity. GWAS is based on linkage disequilibrium for the identification of agronomic traits in a certain crop populations, and has significant advantages, such as high-throughput efficiency, high precision, and less time consumption. Furthermore, it plays a prominent role in crop genetics and breeding [38,39]. Since the successful association analysis of sea beets in 2001 [40], there have been increasing reports on GWAS in *A. thaliana*, rice, rape, wheat, and other crops. By re-sequencing and conducting genome-wide association analysis of 991 accessions worldwide, Wu et al. identified several candidate flowering genes in spring, semi-winter, and winter rapeseed varieties [41]. Raman et al. used 188 rapeseed germplasms to conduct GWAS at the flowering stage in different environments, and identified 69 single-nucleotide polymorphism (SNP) markers related to flowering time, several of which were located in the candidate regions of flowering genes, such as *FT*, *FLC*, *FRIGIDA* (*FRI*), and *CO* [42]. Liu et al. conducted a GWAS of 529 rice varieties at the flowering stage in three environments and found hundreds of important flowering loci that contained new candidate genes along with most known flowering genes [43]. GWAS has been extensively used in the study of candidate flowering genes. Furthermore, several candidate flowering genes have been mapped, which has made important contributions to the study of flowering candidate genes and the genetic improvement of crops.

Rapeseed (*Brassica napus* L.) is a relatively new allopolyploid crop derived from the hybridization of *B. rapa* and *B. oleracea*, and most orthologous genes are functionally redundant [44]. As the characterization of a single gene is complex, studying gene functions requires the simultaneous mutation of all copies of the gene. The development of functional genomics has promoted the progress in gene editing technology and accelerated the pace of reverse genetics. Recently, a CRISPR/Cas9 system that can accurately edit target sites in all biological genomes has been established [45,46]. The CRISPR/Cas9 technology has opened a new era of gene editing, owing to its simple procedure, low cost, and high efficiency, and has been successfully employed in studies with animals and plants [47,48], including wheat, potato, and cotton, suggesting that the system can edit multiple copies of genes in polyploid crops [49,50,51]. To date, CRISPR/Cas9 has been extensively applied in directional mutations in rapeseed, which increased pod shattering resistance, seed yield, and disease resistance, and improved plant structure [52,53,54,55].

To our knowledge, this is the first report to identify the MICK family in *B. napus* and to characterize its domain, motif, and gene structure. In these analyses, combined with a genome-wide association analysis at the flowering stage, *BnaSVP* and *BnaSEP1* were found to be closely related to flowering. Therefore, we obtained *BnaSVP* and *BnaSEP1* mutants with stable genetic backgrounds in rapeseed using the CRISPR/Cas9 system. We found that copies of *BnaSVP* and *BnaSEP1* have functional redundancy, and *BnaSVP* inhibits rapeseed flowering by inhibiting the expression of *BnaFT* and *BnaSOC1*. Mutant plants without T-DNA insertions were obtained in their next generation. Further, the *svp* mutants displayed an early flowering phenotype, ultimately creating various mutant types of early flowering and making important contributions to the breed varieties of early flowering.

## 2. Results

### 2.1. Identification of Westar MIKC Family Members

Most of the MIKC genes are related to flowering. Thus, in this study, the MIKC family was explored to understand the function of flowering genes in rapeseed. In many flowering plants, the number of MIKC genes ranges between 40 and 70 [56]. Although their phylogenetic distance is far, rice and *A. thaliana* have a similar number of MIKC genes (42 and 39, respectively) [22]. A total of 172 genes were identified in the genome of *B. napus* Westar. Further, the number of MIKC genes in *B. napus* was higher than that in most characterized flowering plant species. This finding was partly due to the tetraploid nature of *B**. napus*. Thirty-nine MIKC genes in *A. thaliana* were homologous to thirty-seven genes in Westar (*At1g65360* and *At1g77080* had no homologous genes), indicating that the MIKC genes in *A. thaliana* and *B. napus* had conserved evolutionary patterns. Further, 35 Westar genes had multiple copies (*At1g65360* and *At1g26310* have only one homologous copy), such as *FLC*, *AP1*, *AP3*, *PISTILLATA* (*PI*), *SEP2*, *SEP3*, *SOC1*, and *SVP*. Among them, the three genes *FLC*, *AGL72*, and *AGL18* had more than nine copies. The TM3 subfamily *AGL72* had 14 homologous copies, *AGL18* had 11 homologous copies, and the FLC-LIKE subfamily *FLC* had 9 homologous copies (Appendix A). These genes are either the key genes in the ABCDE model involved in flower morphogenesis (such as *AGL5*, *AP1*, *PI*, and *SEP*), or key genes regulating flowering time (such as *FLC*, *SOC1*, and *SVP*).

To elucidate the evolutionary relationship of the MIKC genes, a phylogenetic tree was constructed using the MIKC genes found in Westar (172 genes) and *A. thaliana* (39 genes) (Figure 1). According to the floral development ABCDE model, 211 members are divided into six categories: A, B, C, D, E, and H [57]. Of these categories, Class A had 11 Westar genes and 3 *A. thaliana* genes (*AP1*, *FUL*, *CAULIFLOWER* (*CAL*)); Class B had 10 Westar genes and 2 *A. thaliana* genes (*PI*, *AP3*); Class C had 4 Westar genes and 1 *A. thaliana* gene (*AG*); Class D had 12 Westar genes and 3 *A. thaliana* genes (*SEEDSTICK* (*STK*), *SHATTERPROOF 1* (*SHP1*), *SHP2*); Class E had 20 Westar genes and 4 *A. thaliana* genes (*SEP3*, *SEP1*, *SEP4*, and *SEP2*); and Class H had 115 Westar genes and 26 *A. thaliana* genes. The H family can be divided into TM3, FLC-LIKE, STMADS11, AGL6, AGL15, and AGL17 (Appendix A). Most members of the TM3, FLC-LIKE, and STMADS11 subfamilies are important for flowering regulation. For example, *SOC1* in the TM3 subfamily is a key gene in this regulatory pathway, *FLC* in the FLC-LIKE subfamily is the central gene in the vernalization pathway, and *SVP* in the STMADS11 subfamily is an important gene in the temperature pathway. The phylogenetic tree showed that the homologous copies of *A. thaliana* and *B. napus* genes could be clustered in the same evolutionary branch, such as *At5g60910* (*FUL*) in the class A gene and five homologous copies in *B**. napus*, *At5g20240* (*PI*) in the class B gene, and six homologous copies in *B**. napus* (Figure 1). These findings indicate that the MIKC family members of *A. thaliana* and *B. napus* have evolutionarily conserved functions.

### 2.2. Chromosome Distributions of the BnaMIKC Genes

Chromosome localization of 172 BnaMIKC genes in *B. napus* Westar was performed and visualized using TBtools software (Figure 2). Among the 172 BnaMIKC genes, 83 were distributed in the A subgenome, 86 were distributed in the C subgenome, and 3 were distributed in random chromosomes. The MIKC family genes were distributed relatively evenly in the genome. Among them, most genes (19 genes) were located on chromosome A03, indicating that chromosome length is not proportional to the number of BnaMIKC genes. The number of BnaMIKC genes on chromosomes A03 and C03 was the highest, with 19 and 16 genes, respectively. Further, the number of BnaMIKC genes on chromosomes A01 and C01 was the lowest (4 genes). The distribution of MIKC family genes on chromosomes was symmetrical and uniform, except on chromosomes A04 and C04. For example, the number of BnaMIKC genes on chromosomes A02 and CO2 was 11 and 12, respectively; the number of BnaMIKC genes on the A05 and C05 chromosomes was 5 and 4, respectively; the number of BnaMIKC genes on the A06 and C06 chromosomes was 6 and 7, respectively; and the number of BnaMIKC genes on the A08 and C08 chromosomes was 6 and 7, respectively (Figure 2).

### 2.3. Protein Domain, Motif, and Gene Structure Analysis

To identify the domains of the MIKC family proteins in *B. napus* Westar, the protein sequences of 172 genes were submitted to NCBI for domain prediction, and the predicted data were submitted to TBtools software for visualization (Figure 3). A total of 172 genes contained the following six conserved domains: MADS, K-box, PEX11, hypF, self-comp, and DUF223. Further, only four genes contained one or two of the following domains: PEX11, hypF, self-incomp, and DUF223 (*BnaC03T0235500WE* contained self-incomp and DUF223, and *BnaC09T0283000WE* contained the hypF domain; both genes were homologous copies of STK and clustered in the same branch. *BnaA04T0011600WE* and *BnaC04T0263900WE* contained the PEX11 domain and clustered in the branches of the AGL6 subfamily). The domains of the MIKC family proteins are crucial for their function. In fact, the MADS domain and K-box domain are indispensable for DNA-binding and protein complex formation, respectively [12]. A total of 157 BnaMIKC genes were encoded for MADS and K-box (91.28%), while only 15 genes lacked the K-box domain (8.72%, Figure 3). It can be seen that MADS and K-box appear simultaneously in most genes in this family.

The 10 motifs of the MIKC family genes were analyzed using the online website MEME, and the motifs and gene structures of the MIKC family genes were jointly constructed into an evolutionary tree using TBtools and MEGA 7 (Figure 4). The highly conserved motifs in 172 genes were motifs 1, 2, 3, 4, 5, and 7. Some subfamily members did not contain these six conserved motifs. In fact, 10 homologous copies of *AP3* in class B did not contain motif 5, and 6 homologous copies of *PI* in class B did not contain motif 4. Some genes also contained specific motifs. For example, members of the TM3 subfamily in class H contain motif 10 specifically, and motif 8 mainly appears in class E genes, which may cause gene function differentiation. The SMART website was used to annotate the motifs; the MADS domain corresponded to motifs 1 and motif 2, while the K-box domain corresponded to motif 3.

Most members had significant sequence identities and similar exon–intron structures within the same subfamily, indicating close evolutionary relationships. Most genes contained 6–7 exons and 5–6 introns. However, *BnaC03T0614700WE*, in the AG subfamily, contained 11 exons, whereas *BnaC03T0107700WE*, *BnaA03T0159800WE*, *BnaA08T0208400WE*, and *BnaA05T0144400WE* contained only 1 exon (Figure 4).

### 2.4. Design of sgRNAs to Knock Out Homologs of the SVP and SEP1 Genes in B. napus

To understand the genetic regulation of flowering time, GWAS was performed on flowering traits. A total of 25 flowering candidate genes were identified. These genes are distributed on nine chromosomes and originated from different flowering regulatory pathways [58]. *BnaA10.FLC*, *BnaA09.SVP*, and *BnaC09.SEP1* are also members of the MIKC family. *FLC* is a vernalization pathway gene, and Westar is a material that lacks vernalization. Therefore, we employed CRISPR/Cas9 to explore the functions of *SVP* and *SEP1* in *B**. napus*.

Studies have shown that *SVP* gene function is crucial for the regulation of flowering time. For example, *SVP* mutants showed early flowering traits in *A. thaliana* [59]. Thus, *SVP* is an ideal gene for creating early flowering varieties of rapeseed. *B. napus* contained four *SVP* copies, namely, *BnaA09.SVP* (*BnaA09g42480D*), *BnaC08.SVP* (*BnaC08g34920D*), *BnaA04.SVP* (*BnaA04g12990D*), and *BnaC04.SVP* (*BnaC04g35060D*). For *B. napus SVP*, the amino acid sequences of *BnaA09.SVP*/*BnaC08.SVP*/*BnaC04.SVP* contained MADS and K-box, whereas *BnaA04.SVP* contained only the K-box domain. The amino acid sequence of *B**naSVP* is highly conserved; the highest amino acid sequence similarity was 99.15% of *BnaA04.SVP* and *BnaC04.SVP*, and the lowest was 83.27% of *BnaC08.SVP* and *BnaC04.SVP* (Appendix A). In cruciferous crops (*B. napus*, *B. rapa*, *B. oleracea*, *A. thaliana*), the MADS domain had no amino acid difference, and the K-box domain had only a few amino acid differences (Appendix A). *BnaC09.SEP1*/*BnaA10.SEP1* contained MADS and K-box, with only one amino acid difference in MADS and two amino acid differences in K-box in *A. thaliana* (Appendix A).

Four copies of *SVP* were present in *B. rapa* and *B. oleracea*. Further, phylogenetic analysis revealed that *BnaA09.SVP*/*BnaA04.SVP* and *BnaC08.SVP*/*BnaC04.SVP* were closely related to their homologs in *B. rapa* and *B. oleracea*, respectively (Appendix A), which is consistent with their origin. *BnaC09.SEP1* and *B**. carinata* clustered in the same branch, whereas *BnaA10.SEP1* and *B. rapa* clustered in the same branch (Appendix A).

Two sgRNAs were used to generate knockout mutations in four copies of *BnaSVP* and two copies of *BnaSEP1*. sgRNA-1 targets exon 8 of *BnaSVP* (downstream of the K-box domain), while sgRNA-2 targets exon 4 of *BnaSVP* (the K-box domain). Both sgRNA-1 and sgRNA-2 could target four copies of *BnaSVP* to avoid functional redundancy between each copy (Figure 5b). Similar to the sgRNA design method in *BnaSVP*, the sgRNA in *BnaSEP1* could target two copies of *BnaSEP1* (sgRNA-1 target exon 6, sgRNA-2 target exon 4; Figure 5c). A CRISPRs/Cas9 vector driven by the 35S promoter (35SP) was used (Figure 5a).

### 2.5. Identification of BnaSVP and BnaSEP1 Mutations

We obtained 62 *BnaSVP* and 22 *BnaSEP1* transgenic plants using tissue culture techniques. PCR analysis revealed that 91.94% (57/62) and 86.36% (19/22) of the T0 lines carried the Cas9 protein (Appendix A).

Mutations in the T0-generation positive transgenic plants were detected using high-throughput tracking of mutation technology (Hi-TOM) [60]. The T0 transgenic plants were found to have high editing efficiency (*BnaSVP*: 90.63%; *BnaSEP1*: 63.16%), and were more likely to produce mutations below three bases (insertion and deletion), with low base-substitution efficiency (Appendix A; Appendix A). Further, the mutation frequency of the A/T base pair was markedly higher than that of the C/G base pair (Appendix A).

The T0 generation was planted in a greenhouse as well as outdoors, and the flowering phenotypes of the T1-generation mutants were investigated. Finally, the *svp* mutant displayed an early flowering phenotype, whereas the *sep1* mutant did not display any phenotypic changes.

Studies on *A. thaliana* have shown that the *svp* mutations cause early flowering in *A. thaliana* [61]. In the outdoor environment, the flowering period of the *svp* mutant was investigated, and the same phenomenon was found in *B**. napus*. The *svp* mutant displayed an early flowering phenotype (Figure 6a). The flowering time of the *svp* mutant was 107.3 ± 9.48 days, the average flowering time of the wild type was 137.6 ± 4.39 days. This difference in flowering time was found to be significant.

In the greenhouse environment, the *svp* mutants also displayed early flowering. When the flowering time of the *svp* mutant was compared with that of the wild-type Westar and transgenic-negative plants, a significant difference was found (Figure 6b–d). According to the statistics of flowering time for each individual plant, the flowering time of the wild type was 83.6 ± 3.78 days, and the flowering time difference between the *svp* mutant and the wild type was 8–31 days. Editing detection and statistics for the T1-generation mutants were generated to determine the reasons for the differences in the flowering period. Homozygous mutations at the double targets of all homologous copies were found to have the greatest impact on flowering differences (such as the SVP-5 and SVP-7 strains). When heterozygous mutations occur, the difference in the flowering period decreases (e.g., SVP-W1-2 strain and SVP-d-5-5). However, when only two homologous copies were edited (e.g., SVP-1-5), no difference in the flowering period was found relative to that of the wild type (Table 1).

### 2.6. Expression Analysis of the BnaSVP Gene

Liu et al. established an online transcriptome platform for the oil crop *B**. napus* [62]. We analyzed the expression pattern of *BnaSVP* on the transcriptome platform (http://yanglab.hzau.edu.cn/BnTIR, last accessed date 28 February 2022) and found that four copies of *BnaSVP* had transcripts. However, different copies of *BnaSVP* had different transcripts in all tissues. *BnaSVP* had a significantly higher expression level in roots, stems, leaves, and vegetative rosettes than in the buds, petals, pollen, siliques, and seeds. Moreover, *BnaC04.SVP* had a higher expression level than the other three copies. These findings indicate that the *BnaSVP* copies play an important role in flowering regulation (Appendix A).

To determine the effect of *BnaSVP* copies mutation on flowering time, four tissues, including the flower, bud, leaf, and stem, of each strain during the same period were selected, and four tissues of wild-type Westar during the same period were selected as controls. Quantitative primers for the *BnaSVP* copies were designed, and qRT-PCR was performed (Appendix A). Based on these results, the expression levels of the *BnaSVP* copies in the four tissues were significantly lower in the experimental strains than those in the wild type, and the decreases in *BnaA09.SVP* and *BnaC08.SVP* were greater than those in *BnaA04.SVP* and *BnaC04.SVP* (Figure 7). *SVP* is an inhibitor of flowering time. The expression levels of *BnaSVP* copies in the mutant decreased, resulting in a weakening of its inhibitory effect on flowering.

By exploring *SVP* in Chinese cabbage, Lee et al. found that *SVP* delayed flowering by inhibiting the expression of *FT* and *SOC1* [63]. To identify additional *SVP* functions, Andres et al. performed genome-wide transcriptome data analysis and found that the expression level of *GIBBERELLIN 20 OXIDASE 2* (*GA20OX2*) was upregulated in *svp* mutants. *GA20OX2* encodes gibberellin (GA). Notably, GA is known to promote flowering [64]. Therefore, we carried out fluorescence quantitative analysis of *FT*, *SOC1*, and *GA20OX2* in the *SVP* mutant to identify the genes that play a role in the promotion of flowering in *B. napus* after *SVP* mutation. The results showed that the expression of *SOC1* was significantly higher in the buds (Figure 8a,b). The expression of *FT* increased significantly in both leaves and buds (increased dozens of times; Figure 8c). Furthermore, the expression of ga20ox2 slightly changed (Figure 8d). Overall, *SOC1* and *FT* were found to play a role in promoting flowering after *BnaSVP* mutation. Higher expression levels of *FT*, *SOC1*, and *GA20OX2* in SVP-7-2 buds corresponded to shorter flowering times.

## 3. Discussion

### 3.1. MIKC Family Genes Provide the Basis for Rapeseed Development

MIKC family genes are of great significance for the development and crop-improvement of rapeseed; however, this family has not been identified in rapeseed, and its function is unknown. Our data show that the MIKC family is a very detailed and large family containing 172 genes. There are 39 MIKC genes in *A. thaliana* [20] and 201 MIKC genes in wheat [65], indicating that the number of MIKC genes correlates with the number of chromosomes. We divided the MIKC family genes into 6 groups according to the phylogenetic tree, and some articles divided them into 14 [66] or 15 subfamilies [65]. Phylogenetic tree and family gene feature analyses showed that their family members had high homology with *A. thaliana* genes, and the gene structures and motifs of each subclass were similar. We selected *BnaSVP* and *BnaSEP1* from this family for functional verification and created early flowering *svp* mutants. Many gene functions in this family are unknown in *B. napus*, and have the value of further research. The identification of MIKC family genes provides a basis for the study of gene function in breeding and the determination of gene-editing objectives to improve the performance of rapeseed.

### 3.2. Homologous Copies of BnaSVP and BnaSEP1 Have Functional Redundancy

Redundancy is an inherent characteristic of organisms that is gradually formed in the process of long-term adaptation to environmental evolution, and gene function redundancy is an important feature for maintaining gene function stability. *B. napus* contains two different subgenomes, but the genetic relationship is closer; therefore, in these two subgenomes of *B. napus*, some important traits may contain copies of gene function redundancy [52,67]. When only *BnaA09.SVP* was homozygously mutated, the flowering time did not change, but only when all four copies of *BnaSVP* were mutated, the flowering time changed most significantly (Table 1), indicating that the four copies of *BnaSVP* are functionally redundant. The tissue-specific expression of *BnaSVP* showed that the expression level of each copy was different, so we speculate that although each copy was functionally redundant, its contribution to flowering time regulation may be different; each copy needs to be mutated to further validate this conjecture. Moreover, the expression level of *BnaSVP* changed significantly during the transition from vegetative growth to reproductive growth (Appendix A), implying that *BnaSVP* is a key node gene involved in the regulation of flowering in rapeseed, which is consistent with our experimental results (Figure 6; rapeseed showed early flowering after *BnaSVP* mutation).

In this study, we constructed a CRISPR/Cas9 vector to knock out *BnaC09.SEP1* and *BnaA10.SEP1*, and the mutants did not show new phenotypes. When the Westar reference genome was published, we aligned *BnaC09.SEP1* to the genome and found that *BnaC09.SEP1* has three homologous copies (*BnaA10.SEP1*, *BnaA03.SEP1*, *BnaC03.SEP1*) [68]. We designed specific sgRNA that only mutated *BnaC09.SEP1* and *BnaA10.SEP1*; therefore, we speculated that the reason for the absence of phenotypic changes in the *BnaSEP1* mutants may be that the copies of *BnaSEP1* are functionally redundant, and copies without mutations restore the phenotype (*BnaA03.SEP1* and *BnaC03.SEP1*). Therefore, redundancy between gene copies should be considered when studying gene function. Using CRISPR/Cas9 to simultaneously mutate multiple genes plays an important role in studying gene function and in producing important traits.

### 3.3. CRISPR/Cas9-Targeted Mutation in BnaSVP Is a Promising Strategy for Early Rapeseed Breeding

We analyzed the inheritance law of the mutation sites from the T0 generation to the T2 generation. Accordingly, we found that most mutation sites could be stably inherited between generations, and new mutation sites may appear when the Cas9 protein still exists (Figure 9). Sun et al. found that many new editing events appeared in the T1-generation plants [55], and Yang et al. also found new mutation types in the T1 generation, and believed that they might be the result of Cas9 continuing to play a role in the non-mutated allele of the target region [69]. Furthermore, the mutant plants lost T-DNA between generations (11.1% *SVP* (8/72) and 18.2% *SEP1* (4/22)) (Appendix A). Ideal plants without T-DNA were obtained by genetic isolation from rapeseed, which can promote the development of biological breeding and avoid the commercialization of transgenic plants [70].

The CRISPR/Cas9 system has potential off-target effects, but numerous studies have shown that off-target effects can be ignored when sgRNAs have high specificity. The CRISPR/Cas9 system has good specificity for targeted mutations of *B. napus*, which can accurately mutate the genome sequence of *B. napus* [52,54,67,69,71]. Moreover, unnecessary off-target effects in plants can be eliminated by hybridization with parents [72]. Therefore, off-target events were not considered in this study.

Research on flowering genes in *A. thaliana* has revealed several important details; however, research on flowering genes in *B. napus* remains insufficient. Flowering is of great significance to crop yield and production. Early maturity and high yield are the main breeding objectives for the early breeding of rapeseed [73]. Previous studies have shown that early flowering correlates with early maturity. Rapeseed has a long growth period, and crop rotation is often carried out with rice and cotton, resulting in a large reduction in the planting area. Breeding early maturing varieties is the key to solving the problem of crop rotation. Furthermore, the adverse natural environment could be avoided to increase the yield. Therefore, the use of a rapid genetic variation to improve this trait is crucial. As a result, a rapid and accurate technology is needed to generate mutations simultaneously in all homologs of *SVP* and *SEP1* in *B. napus*. CRISPR/Cas9 technology can achieve this goal while simultaneously creating multiple gene mutations. To date, several new traits were obtained in rapeseed by this technique, such as self-incompatibility and seed coat color, through the generation of specific gene knockouts [67,74]. However, studies on early flowering gene mutants in rapeseed have rarely been reported.

In this study, we used CRISPR/Cas9 to effectively mutate *BnaSVP* gene in *B. napus*. An early flowering phenotype was observed only when four functional copies of *BnaSVP* mutated simultaneously (Figure 6). This finding further supports the notion that *SVP* is a key gene regulating flowering time and is highly conserved in Brassica species. Quantitative fluorescence analysis of the *SVP* gene and its related genes in the mutant further confirmed that the early flowering time of the mutant was due to decreased *SVP* levels, which led to an increase in *FT* and *SOC1* levels and promoted flowering, indicating that *BnaSVP* inhibited *B. napus* flowering by inhibiting the expression of *FT* and *SOC1* (Figure 7 and Figure 8). The greatest mutational effect occurs when all four copies of SVP are homozygously mutated; the mutant flowering time was approximately 30 days earlier (Table 1). Therefore, the mutants obtained in this study will provide valuable materials for the early flowering and early maturation breeding of rapeseed.

### 3.4. SVP Gene Has Different Functions

*SVP* has the same function of inhibiting flowering in *Chrysanthemum morifolium* [75] and trifoliate orange [76] as in *A. thaliana*. However, ectopic expression of soybean *SVP*-like genes in tobacco showed the opposite result (early flowering) [77]. In *Medicago truncatula* (*Medicago*) [78] and perennial ryegrass [79], *SVP* genes do not regulate flowering time. We analyzed the domain structure of *SVP* in these crops and found that trifoliate orange, *Chrysanthemum morifolium*, soybean, and *Medicago truncatula* contained MEF2-like MADS and K-box domains, and their structures were consistent with SVP in *B. napus*, indicating that the functional differentiation of *SVP* in different crops was not caused by domain structure. Zhang et al. found that different positions of ATG-binding exons in soybean resulted in differences in transcriptional regulation (gene function changes, early flowering) [77]; Jaudal et al. believed that there was a lack of SVP protein interaction partners (such as *FLC*) in *Medicago truncatula*, so the *SVP* gene could not regulate the flowering time of *Medicago* [78]. *SVP* is not only related to flowering time. In fact, a natural variation of *SVP* contributes to elongated glumes in *Triticum petropavlovskyi* [80]. The *SVP* gene also participates in the dormancy of eudicots [81]. Overall, these findings emphasize that the *SVP* gene has multiple important functions and research values.

In this study, we temporarily observed only that *SVP* can regulate the flowering time of rapeseed. Further, no other functions were found. We predicted the protein interaction network of *SVP* and found that it mainly interacted with the *AP1* and *FT* proteins (Figure 10). *SVP* and *AP1* controlled flower development and regulated the expression of class B, C, and E genes. *SVP* and *FT* jointly regulate flowering time. Lee et al. found that *SVP* binds to the vCArGIII motif in *FT* promoter to inhibit *FT* function in *A. thaliana* [82], which is consistent with our protein network and experimental results. When the inhibitory effect of *SVP* on flowering was weakened, the expression of *FT* increased (Figure 6c), and the flowering time of the mutant advanced.

## 4. Materials and Methods

### 4.1. Plant Materials

Westar is the genetic transformation material of *B. napus*. Westar is a spring rapeseed without vernalization that possesses a high transformation efficiency, a feature that led to its selection as the experimental material for this study.

### 4.2. Identification of MICK Family Members in B. napus

The sequence information of Westar was downloaded from the *B. napus* genome information resource network (http://cbi.hzau.edu.cn/cgi-bin/rape/, last accessed date 28 February 2022). BLASTP by known *A. thaliana* protein sequences, and an HMMER3 search was performed to screen candidate genes [83]. Thereafter, the protein sequence was submitted to the NCBI conserved domain database (CDD, https://www.ncbi.nlm.nih.gov/cdd, last accessed date 28 February 2022) and SMART database (http://smart. embl-heidelberg.de/, last accessed date 28 February 2022). The integrity of the MADS domain was confirmed, and genes without a complete conserved MADS domain were deleted [84,85].

### 4.3. Analysis of MICK Family Gene Characteristics and SVP Protein Interaction Network

ClustalW was used to carry out multiple sequence alignments for the MICK family proteins in *B. napus* and *A. thaliana*. The phylogenetic tree was constructed by MEGA 7, and beautified using the online website EvolView (https://evolgenius.info/, last accessed date 28 February 2022). The motif and domain were identified using MEME (http://meme-suite.org/tools/meme, last accessed date 28 February 2022) [86] and CDD, respectively. Finally, the gene structure, domain, and motif were visualized using TBtools [87]. STRING11 tool [88] was used to predict the *SVP* protein interaction network.

### 4.4. CRISPR/Cas9 Vector Construction and Plant Transformation

The CRISPR/Cas9 editing vector was constructed using an online website to design sequence-specific sgRNAs (http://crispr.hzau.edu.cn/cgi-bin/CRISPR2/CRISPR, last accessed date 28 February 2022). The vector was pKSE401 (kanamycin resistance), which could edit multiple genes simultaneously, and the vector was constructed according to the method provided by Xing et al. [89]. Sequencing was performed to verify whether sgRNAs were successfully assembled with the vector, using the dark–light culture method and the transformation of rapeseed hypocotyl by *Agrobacterium tumefaciens* [90].

### 4.5. Positive Transgenic-Plant and Mutant Identification

Positive plants were judged by detecting the presence of cas9 in transgenic plants; thereafter, cas9 protein was amplified using the specific primer cas9-F/R (Appendix A). The mutation sites of transgenic plants were detected by high-throughput sequencing, and the sequencing data were submitted to the Hi-TOM platform to visualize the mutation sites [60]. Amplicons were sequenced after two rounds of PCR as follows: (1) using specific primers to amplify target sites (the first round of PCR, Appendix A); (2) using universal primers to mark each single plant (the second round of PCR); (3) mixing the second round of amplification products for sequencing (Novogene Bioinformatics Institute, Beijing, China); (4) decoding the mutation sequence on the online Hi-TOM website (http://www.hi-tom.net/hi-tom/, last accessed date 28 February 2022) to determine the mutation type of each plant at each target.

### 4.6. Phenotypic Observations

CRISPR/Cas9 transgenic plants were transplanted into the greenhouse from M4 culture medium at a greenhouse temperature of 22 °C and a photoperiod of 16 h light/8 h dark. Positive and mutation detection of the T0-generation plants was carried out in the greenhouse. Mutant plants of the T1 generation were planted in both outdoor and indoor environments, and 12–20 plants were planted in each line for phenotypic observation. The transgenic cultivation base at Wuhan Huazhong Agricultural University served as the outdoor environment. The time period between sowing and the first flower’s opening was defined as the standard flowering period of a single plant, and the average flowering date of a single plant was defined as the flowering period of the lines.

### 4.7. RNA Extraction and Quantitative Real-Time PCR (qRT-PCR)

RNA was extracted from stems, leaves, buds, and flowers. Total RNA was extracted from the samples using a polysaccharide polyphenol plant total RNA extraction kit (Tiangen Biotech, Beijing, China). A reverse transcription kit (Prime ScriptTMRT reagent Kit, TaKaRa, Beijing, China) was used for reverse transcription of the extracted RNA, and qRT-PCR was performed after the cDNA was diluted 50 times. In qRT-PCR, specific primers that distinguish each homologous copy were designed, and reference genes were used as controls (Appendix A). Specific primers were designed using SnapGene software, and the amplification length was between 100 bp and 250 bp. First, the CDS sequence of a gene was aligned with other copies of CDS and DNA sequences. A primer was designed in the region with base differences, and exon spanning primers were designed to avoid the amplification of genomic DNA. Second, the designed primers were submitted to the GENEBANK website (http://www.ncbi.nlm.nih.gov/genbank/, last accessed date 28 February 2022) to detect primer specificity using 2 × SYBR^®^Green Real time PCR Master Mix for qRT-PCR. The relative expression of genes was analyzed according to the 2^−ΔΔCt^ method [91]. GraphPad Prism 8 software was used to visualize the data, and the expression levels between samples were analyzed using a paired two-tailed *t* test.

## Figures and Tables

**Figure 1 ijms-23-04289-f001:**
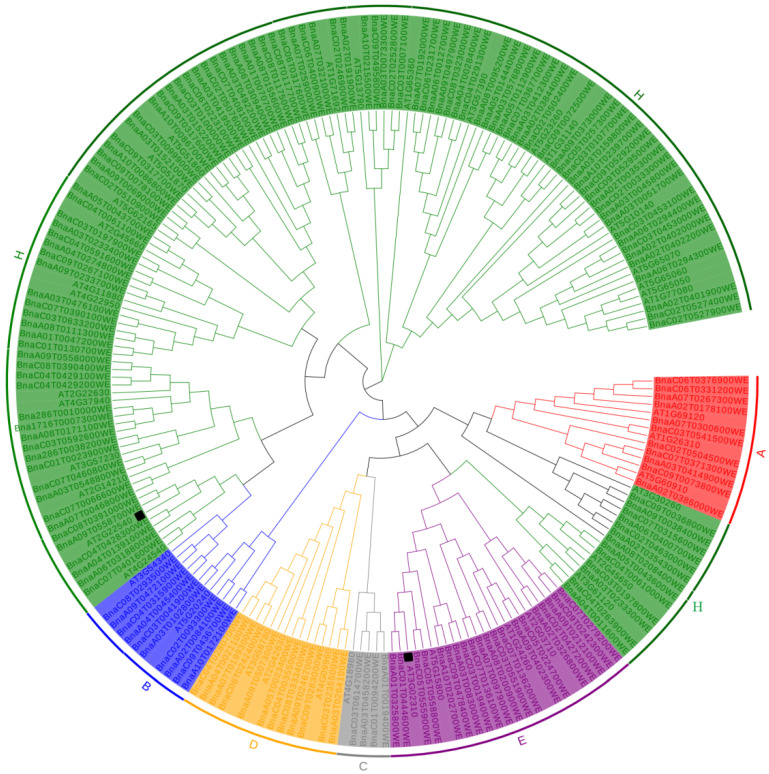
Phylogeny of the MIKC proteins from *B*. *napus* and *A. thaliana*. HMMER and BLASTP were used to identify MIKC genes [24].

**Figure 2 ijms-23-04289-f002:**
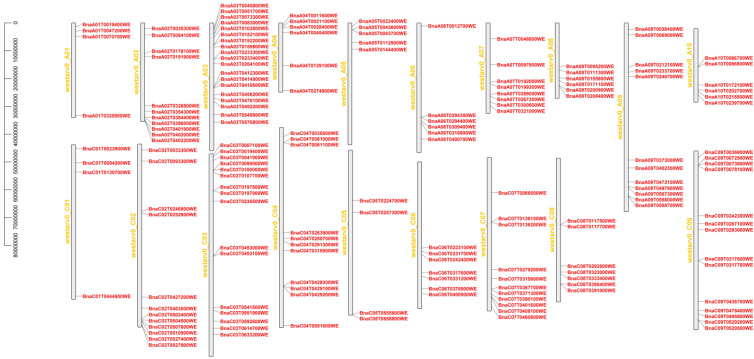
Chromosomal distribution of *B**. napus* BnaMIKC genes. The scale represents base pairs (bps). The chromosome numbers are shown on the left side of each vertical bar.

**Figure 3 ijms-23-04289-f003:**
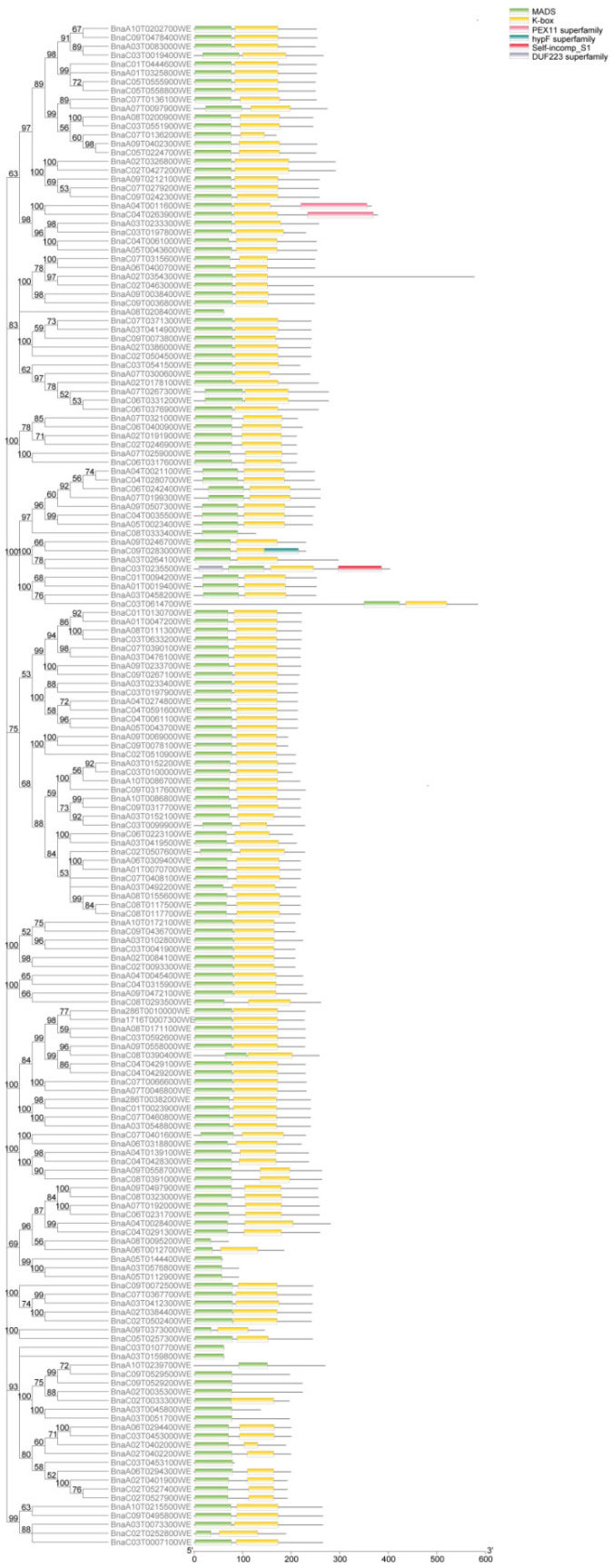
Domain analysis of the MIKC family members.

**Figure 4 ijms-23-04289-f004:**
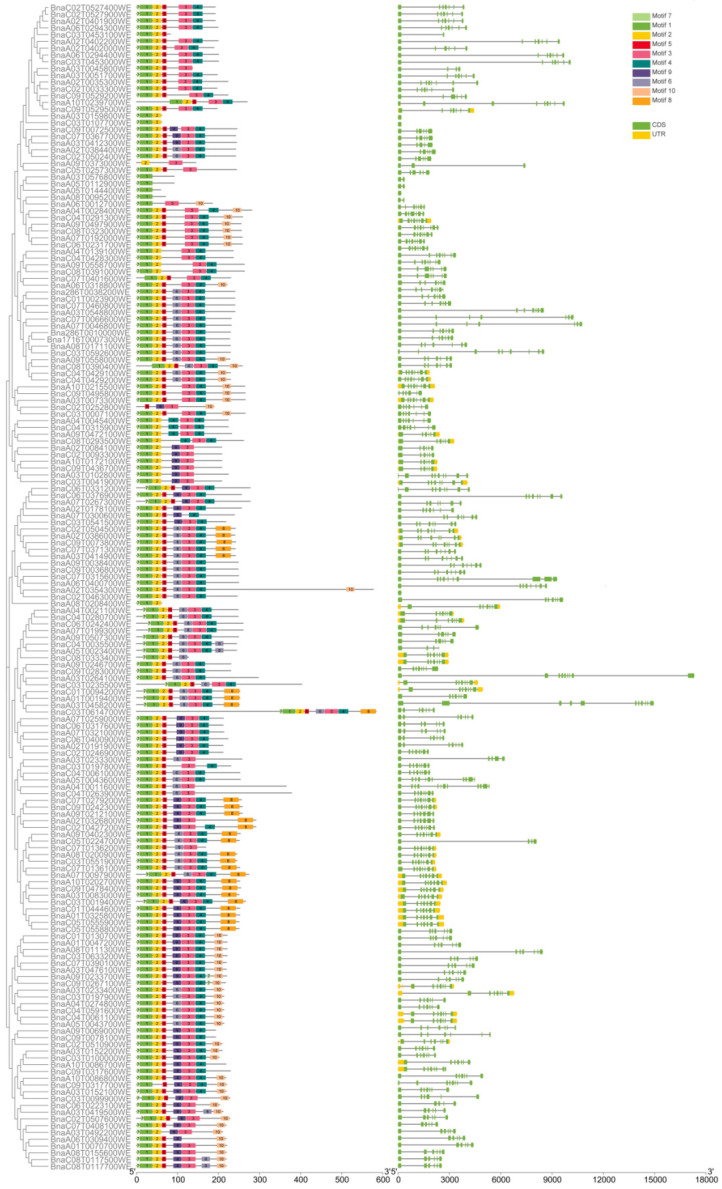
Motif and gene structure analysis of the MIKC family members.

**Figure 5 ijms-23-04289-f005:**
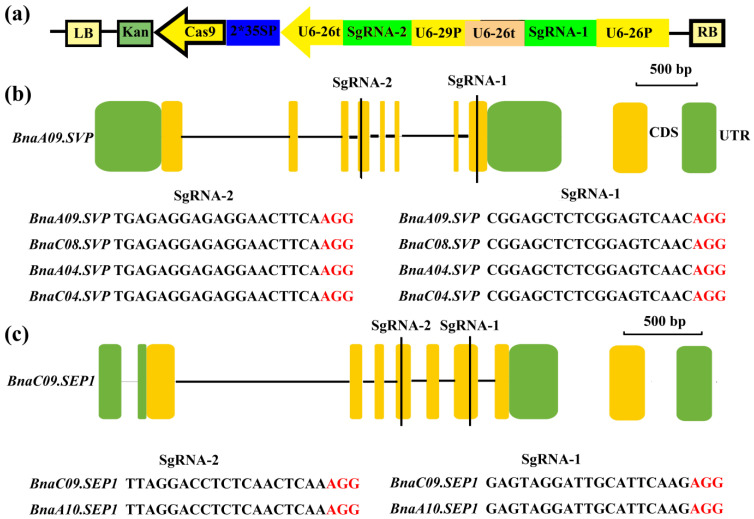
Schematic of *BnaSVP*- and *BnaSEP1*-editing vectors with target sequences. (**a**) Construction of the binary plasmid vectors that house the following: a kanamycin-resistance cassette driven by 35SP; a Cas9-expression cassette driven by 35SP; and U6-26 and U6-29, which can drive two sgRNAs. (**b**,**c**) The single exon of *BnaSVP* and *BnaSEP1* is shown as a yellow box and the single intron is shown as a horizontal line; the black vertical line indicates the knockout position. The target sequences are shown with the PAM sequences (red font).

**Figure 6 ijms-23-04289-f006:**
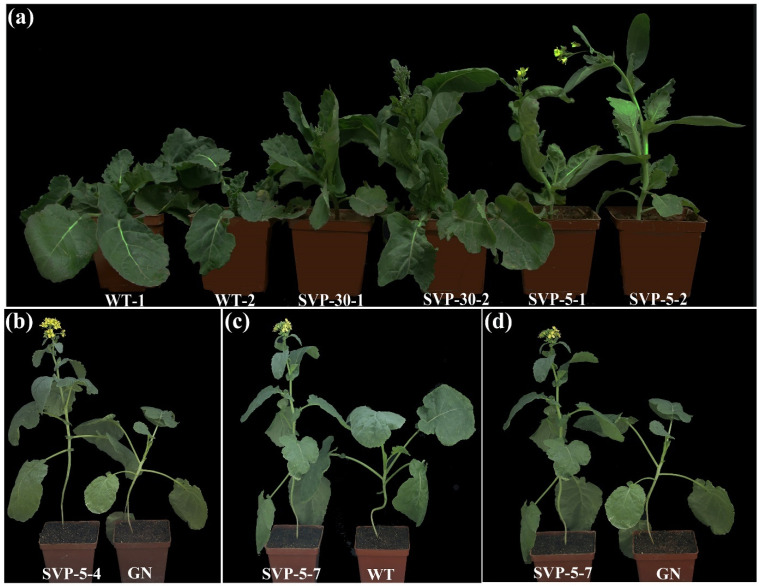
Flowering phenotype of the T1-generation *svp* mutant in outdoor and greenhouse environments. (**a**) The T1-generation *svp* mutant displayed the early flowering phenotype in the outdoor environment. (**b**–**d**) The T1-generation *svp* mutant displayed the early flowering phenotype in the greenhouse environment. WT indicates wild type, and GN indicates transgenic negative.

**Figure 7 ijms-23-04289-f007:**
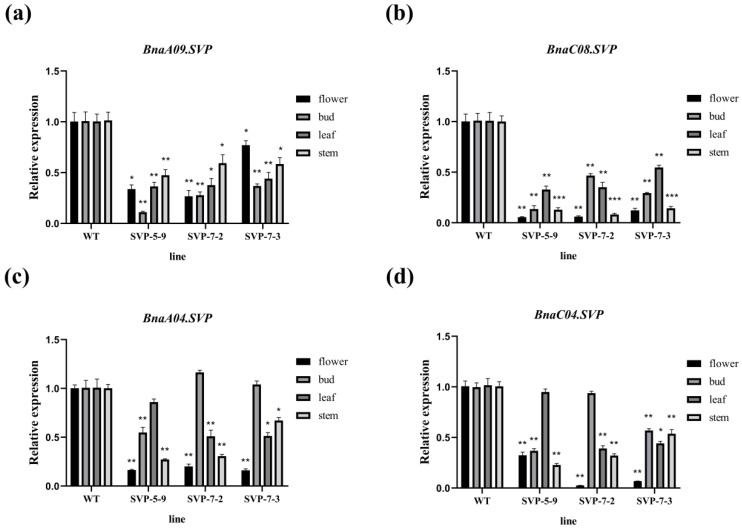
Fluorescence quantitative analysis after *BnaSVP* copy mutation. (**a**–**d**) The expression levels of *BnaA09.SVP*, *BnaC08.SVP*, *BnaA04.SVP*, and *BnaC04.SVP* were altered in the mutants. WT represents wild type; * *p* < 0.05, ** *p* < 0.01, *** *p* < 0.001.

**Figure 8 ijms-23-04289-f008:**
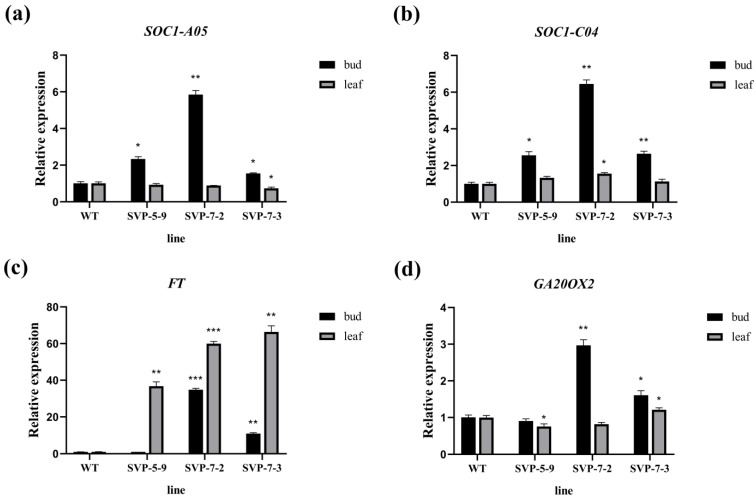
Fluorescence quantitative analysis of the *SVP*-related genes. (**a**–**d**) The relative expression levels of *BnaA05.SOC1*, *BnaC04.SOC1*, *FT*, and *GA20OX2* in the *svp* mutant lines; * *p* < 0.05, ** *p* < 0.01, *** *p* < 0.001.

**Figure 9 ijms-23-04289-f009:**
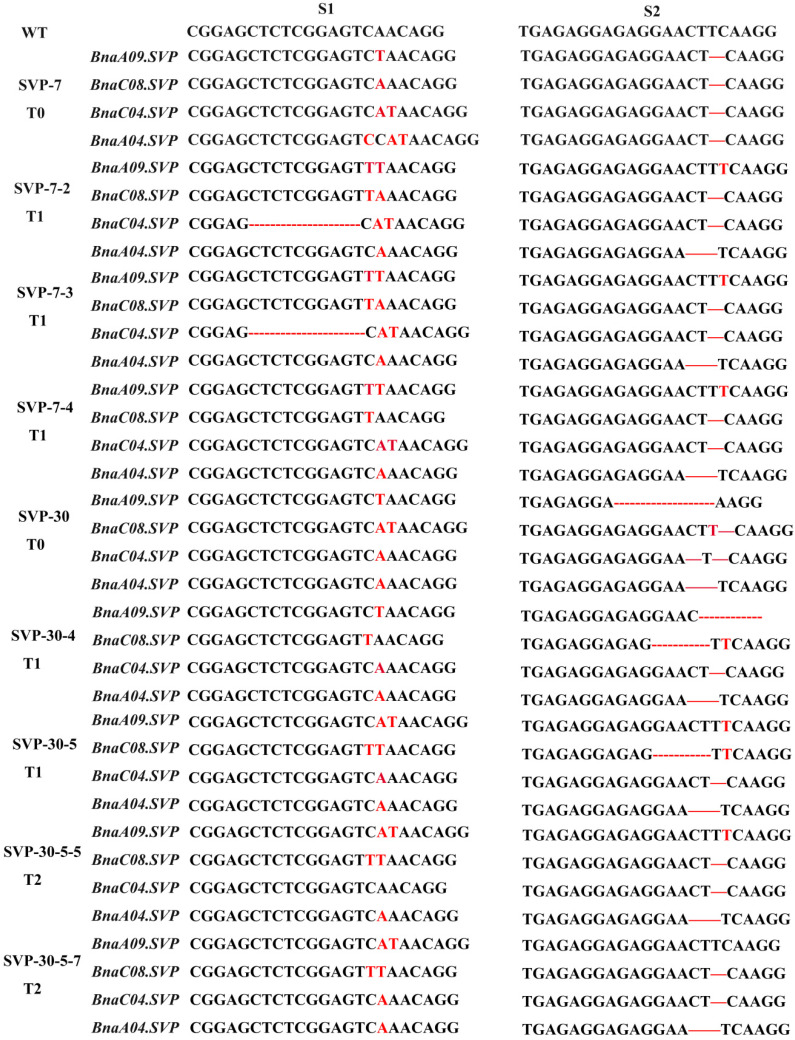
Genetic law of S1 and S2 target sites of *BnaSVP*. Red fonts represent insertion or substitution, red “-” represents deletions. S1 and S2 indicate target site 1 and target site 2, respectively.

**Figure 10 ijms-23-04289-f010:**
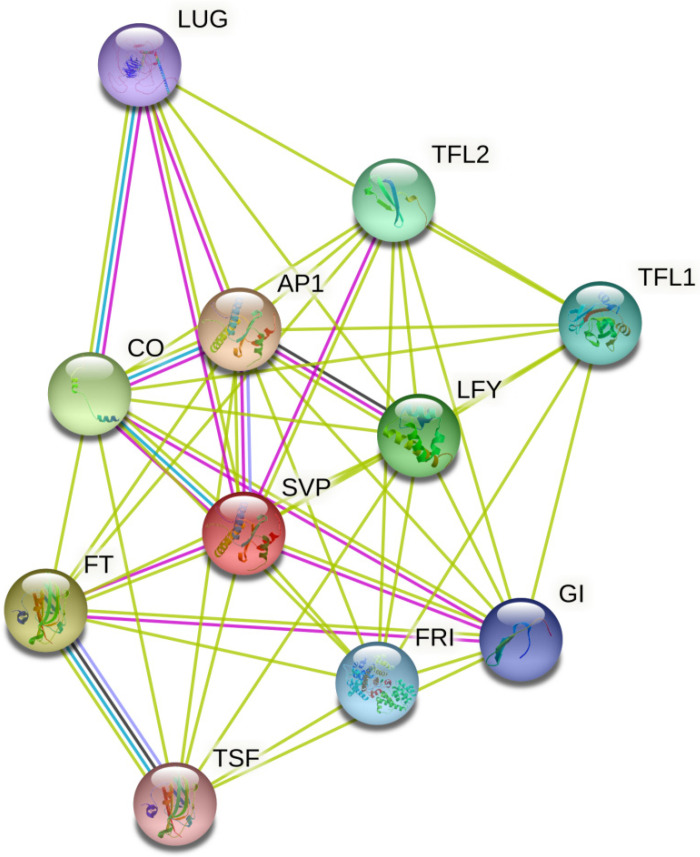
Prediction of the *SVP* protein interaction network. Loading *SVP* gene into String 11.0 online tool for protein interaction network prediction (https://string-db.org/, last accessed date 28 February 2022).

**Table 1 ijms-23-04289-t001:** Analysis of the mutation types and flowering period differences for the T1-generation *svp* mutants.

Gene Target Lines	*BnaA09.SVP*		*BnaC08.SVP*		*BnaC04.SVP*		*BnaA04.SVP*		Flowering Period Difference
	Target 1	Target 2	Target 1	Target 2	Target 1	Target 2	Target 1	Target 2	
SVP-W1-2	I	D^+^, H	I	D	I	D^+^	D	D, H	−20
SVP-W1-4	I	D^+^, H	I	D, H	I	I	I, D, H	I, H	−13
SVP-1-5	D	WT	WT	WT	WT	WT	WT	D, H	3
SVP-5-4	I, D^+^	D	D	I, D^+^	I	I, D	I	D^+^	−31
SVP-5-8	D^+^	D	D	D^+^	I	D	I	D^+^	−31
SVP-5-9	I	D, H	I	I	I	I	I	D^+^	−24
SVP-7-2	I, G->A	I	I, G->A	D	I, D^+^	D	I	D	−29
SVP-7-3	I, G->A	I, H	I, G->A	D	I, D^+^	I	I	D	−23
SVP-7-4	I, G->A	I, H	G->A	D	I	D	I	D	−17
SVP-8-5	WT	WT	WT	WT	WT	WT	WT	WT	−2
SVP-11-3	I	D, H	I	D^+^	I	D	I	D	−21
SVP-12-4	WT	D, I^+,^ H	I	D^+^	WT	D, H	WT	D^+^	−9
SVP-26-2	D^+^, H	D^+^, H	I, H	WT	I	WT	I, H	WT	−9
SVP-26-9	D^+^, H	WT	I, H	WT	I	WT	I, H	WT	−8
SVP-30-5-5	I	I, H	I, G->A	D	WT	D	I	D	−18
SVP-30-5-7	I	WT	I, G->A	D, H	I	D	I	D, H	−13
SVP-D-4-10	I	D	I, H	D	I	I	I, H	I	−8
SVP-D-5-5	I	D	I	D	I, D^+^	I, D^+^	I, H	I	−21

I represents insertion; D represents deletion; WT represents wild type; H represents heterozygous mutation; ^+^ indicates more than 3 bases; -> indicates substitution.

## Data Availability

Not applicable.

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
