# Peer review of "Identification and Characterization of the MIKC-Type MADS-Box Gene Family in Brassica napus and Its Role in Floral Transition"

_ijms, 2022, doi:10.3390/ijms23084289_

Round 1

Reviewer 1 Report

The authors provided comprehensive analyses on identifying, mutating and phenotyping a flowering related gene, SVP, in rapeseed. I feel the results were well presented but lacks some details and discussion about the results. For example, the sequences for BnaA09.SVP, BnaC08.SVP, BnaC04.SP, BnaA04.SVP should be provided to demonstrate and explain technical details such as, how similar are the sequences? and how did the authors discriminate the difference when the mutations were introduced by Cas9? Similarly, how were the primers for qRT-PCR designed to discriminate between these copies? Also, the authors demonstrated successful mutations of the SVP gene copies, but potential off-target by Cas9 was not investigated. Analyses of off-target sites would further support that at least some of the potential off-target sites, especially if there are any conserved regions in the genome, are not mutated, hence not affecting the flowering phenotype. For the Discussion part, the authors discuss about the use of Cas9 for rapeseed and some insights about the function of SVP, however, there are many results that are not discussed and could be deepened. For example, have mutations changing over generations (Figure 9) been reported previously? What is the significance of tissue specific expression change data of SVP in relation to its predicted function? How about the results of SEP1? Why was there no phenotypic change observed in SEP1 mutants? Some minor comments were also listed below.

Minor comments

Line 54: respond -> response

Line 103: Unless there are specific reasons, Zhenjiang University should be referred as the authors name (i.e. Wu et al) to keep the format consistent

Line 131: the font size of “efficient knockouts of” is large

Line 211: Ar. thaliana -> A. thaliana

Line 213: I could not understand what “more than nine homologous copies of three genes” mean? Which genes?

Figure 1: STMADS and H are both colored green. It would be better to represent these in different colors so it is easier to see the difference. There is also another cluster between A and E that is not labeled. What class is this?

Figure 2: The resolution of the picture needs to be improved to be able to read the gene labels.

Line 300-309: I could not follow how the 25 flowering candidate genes were found. The diversity panel including the 523 cultivars and inbred lines in eight different environments were from this study? Are there any data associated with this?

Line 343: 35S promoter and 35SP is used. The expression should be kept consistent.

Line 391: The figure legend seems to be incomplete. “-> indicates”…

Section 4.1 ad 4.2 discuss about Cas9 genome editing technology, which may be better combined.

Line 493-500: The authors discuss about the function of SVP in other plant species, which has a different function to flowering. How about the domain structure of these SVPs compared to the flowering type?

Line 501-509: The authors hypothesize the function of SVP binding to the vCArG III motif in the FT promoter repressing its expression. Is the vCArG II motif also found in Bna FT?

Reviewer 2 Report

L.69. It should be added that MIKC-type has been subdivided into MIКCc and MIКC* groups and MIKCc contains 13 different gene subfamilies (Becker and Theißen, Mol. Phylogenet. Evol. 2003, 29, 464–489; Diaz-Riquelme et al., Plant Physiol. 2009, 149, 354–369; Qu et al., Int. J. Mol. Sci. 2021, 22, 10128). Since these subfamilies are mentioned in subsection 3.1, this information is important.

MICK-type appears repeatedly in the text (L.127, 143, 149, etc.). Correct please.

L. 301. The description of GWAS is missing in the Materials and Methods.

L.434-442. Perhaps this paragraph should be moved to the Introduction.

L.495. medicado should be capitalized, italicized and the plant species indicated.

The Discussion section should be significantly expanded. Now it only contains genome editing, but the Results section also includes identification of MADS-box genes, gene structure analysis, etc.

Ref. 1, 4, 25, etc. – article page numbers are missing. 

Round 2

Reviewer 1 Report

Dear Authors,

I believe the authors have appropriately responded to all of my comments. I have several more questions raised by the response and some minor corrections listed below.

Line 162: sites.[64]. -> remove “.”

Line 166: Please revise this sentence to indicate the “company” for sequencing.

Line 470: “have high specificity,” would read better to close the sentence here “have high specificity.”

Line 468: “plants.[79] -> remove “.”

Line 442: Perhaps, I am misunderstanding something, but I couldn’t find the data to support “its contribution to flowering time regulation was different”. From Table 1 and Figure 7, gene expression difference was seen, but the flowering period difference of SVP -5-9, -7-2, and -7-3 was -23—29, which seems to be relatively similar compared to some of the other lines. Would this be an overstatement? Also, because all three lines have mutations in all four copies of SVP, how is the effect of each copy discriminated? I thought that each copy will need to be mutated independently to demonstrate the function of each copy.

Line 447: I understand that redundancy of the gene copies can complement the function of the mutants. In this case, was the SEP1 sgRNA homology specific to BnaC09.SEP1 and Bna A10.SEP1? I am assuming that the sgRNA was designed in a sequence region specific to these two copies, as it should have mutated other copies if the sgRNA was designed to target the conserved region. I feel that the clarification of this point would support the discussion.
